# Structural and Experimental Study of a Multi-Finger Synergistic Adaptive Humanoid Dexterous Hand

**DOI:** 10.3390/biomimetics10030155

**Published:** 2025-03-03

**Authors:** Shengke Cao, Guanjun Bao, Lufeng Pan, Bangchu Yang, Xuanyi Zhou

**Affiliations:** College of Mechanical Engineering, Zhejiang University of Technology, Hangzhou 330009, China; 2112102136@zjut.edu.cn (S.C.); gjbao@zjut.edu.cn (G.B.); 2111702320@zjut.edu.cn (L.P.); 2111802253@zjut.edu.cn (B.Y.)

**Keywords:** underactuation dexterous hand, synergistic adaptive, pressure sensor, anthropomorphic

## Abstract

As the end-effector of a humanoid robot, the dexterous hand plays a crucial role in the process of robot execution. However, due to the complicated and delicate structure of the human hand, it is difficult to replicate human hand functionality, balancing structural complexity, and cost. To address the problem, the article introduces the design and development of a multi-finger synergistic adaptive humanoid dexterous hand with underactuation flexible articulated fingers and integrated pressure sensors. The proposed hand achieves force feedback control, minimizes actuator use while enabling diverse grasping postures, and demonstrates the capability to handle everyday objects. It combines advanced bionics with innovative design to optimize flexibility, ease of manufacturing, and cost-effectiveness.

## 1. Introduction

The dexterous hand is a versatile robotic end-effector developed for multi-tasking [1]. As a multifunctional intelligent system, it has a wide range of potential uses to replace humans in performing tasks in extreme or hazardous environments, such as underwater energy development, space exploration, disaster rescue, nuclear radiation, and other environments [2,3,4,5].

Dexterous hands are distinctly categorized into two actuation modes: full actuation and underactuation. Full actuation is defined by a one-to-one correspondence between the number of actuators and the degrees of freedom, which can be considerable in manipulators. This approach necessitates high levels of compactness and integration in design, often accompanied by substantial costs [6,7,8]. In contrast, underactuation refers to a system with fewer actuators than the degrees of freedom [9]. Different underactuation dexterous hands are suitable for grasping different forms and types of target objects. Still, all of them utilize the natural adaptive nature of underactuation dexterous hands, which can follow the shape of the grasped object and automatically envelope the object without the need for precise control of each joint of the hand [10,11,12,13,14,15]. The Pisa/IIT hand [11], developed in France in 2014, uses only one motor to drive 19 joints. An upgraded version, the Pisa/IIT2 hand [12] was introduced in 2018, which has two embedded motors to drive the five fingers. However, it can only achieve one grasping posture with extremely limited movements. The Ritsumeikan hand [13], developed in Japan in 2013, contains three motors, one of the motor drives the swing–flexion–extension coupling joint of the thumb, one motor controls the flexion–extension of the index finger, and the other motor controls the flexion–extension of the remaining three fingers to be able to grasp most of the objects, and this drive distribution method is a new way of thinking, which takes into account both power grasping and precision grasping. However, there is no sensing feedback on the hand, and the application scenarios are limited. The Chinese University of Hong Kong designed a BCL-13 hand in 2018 [14], with four fingers and four pneumatic actuators, which is better in terms of suppleness and grasping ability, but the gas drive also makes the control particularly difficult, as well as the anthropomorphism degree is not enough. Currently, underactuated dexterous hands have achieved significant progress in terms of integration and anthropomorphism. However, there remains a considerable gap between the number of actuators and the selection of grasping postures. Designing a highly anthropomorphic and integrated dexterous hand that leverages the principles of underactuation while achieving a balance between the number of actuators and grasping configurations holds great application potential.

In this paper, the human hand is taken as the bionic object, the motion model of the dexterous hand to be designed is constructed by combining robot bionics, and a humanoid dexterous hand with internal integrated sensors is designed by using the underactuation method. The first innovation of this paper is the division of the five fingers into three groups for control based on an analysis of real gripping patterns, enabling two distinct modes: power gripping and precision pinching. The second innovation involves the design of a movable pulley on a specific track as a differential mechanism, which achieves underactuation between different fingers and reduces the required driving force. Finally, the gripping performance of the hand is analyzed and evaluated in this paper.

## 2. Dexterous Hand Structural Design

### 2.1. Overall Design of the Dexterous Hand

#### 2.1.1. Selection of Parameter

The human hand consists of five fingers and a palm. The palm is a flexible base, and most of the operations of the human hand depend on the five dexterous fingers. There are 27 bones in the human hand, including 8 carpal bones, 5 metacarpals, and 14 phalanges, of which the phalanges consist of 5 distal phalanges, 4 intermediate phalanges, and 5 proximal phalanges [16]. The human thumb consists of the distal and proximal phalanges and has three kinematic joints: the phalangeal joint, the metacarpophalangeal joint, and the carpometacarpophalangeal joint, with a total of five degrees of freedom. The remaining four fingers consist of three phalanges, distal, middle, and proximal, with distal phalangeal, proximal phalangeal, and metacarpophalangeal joints, with a total of four degrees of freedom [17,18,19].

According to a comprehensive study of human proportions by the National Aeronautics and Space Administration (NASA) [20], the average hand size of an adult is shown in Table 1. The specific parameters of each finger joint of the dexterous hand are shown in Table 2.

In this paper, the dexterous hand is designed to consider the influence of the joint motion range on the workspace of the dexterous hand, and it is necessary to ensure that the tip of the finger can touch the palm at the bending limit position. Therefore, the motion capability of DIP joints is weaker than that of PIP and MCP joints, and the set joint motion ranges are shown in Table 3. The overall configuration of the dexterous hand is shown in Figure 1.

#### 2.1.2. Selection of Materials

In this paper, the prototype body of the dexterous hand is primarily manufactured using 3D printing. Among the 11 types of 3D printing technologies, fused deposition modeling (FDM) and stereolithography (SLA) are the two most widely used [21,22,23]. Fused deposition modeling involves heating thermoplastic materials and extruding them layer by layer, making it suitable for a variety of thermally stable and corrosion-resistant materials such as ABS, PLA, nylon, and TPU. Stereolithography, on the other hand, cures photosensitive polymers by irradiating them with ultraviolet light and is commonly used to print photosensitive resins with various properties. Compared to FDM, SLA can achieve higher precision, smoother surfaces, and better overall appearance, but it is also more costly and complex to manufacture. The performance of three different printing materials is compared in Table 4.

Due to the low molding precision of PLA, the printed fingers experience significant friction at the joint connections during movement. Additionally, the snap structures on the knuckles are used for mounting sensors. However, these structures are prone to slight deformation during the printing process. This deformation can complicate sensor calibration. Ordinary resin offers high printing precision and a smooth surface, which avoids the problem of excessive friction in joint movement. However, due to its low strength and toughness, parts printed with this material are prone to fracture at weak points. Abrasion-resistant resin, a non-degradable material, features a self-lubricating surface and is suitable for applications involving extrusion and impact. Its precision and strength meet practical requirements, significantly reducing friction generated by finger joints during movement. It also accommodates the design of multiple snap fasteners in the finger while meeting the overall strength requirements. Therefore, considering both performance and cost, this paper selects wear-resistant resin as the material for the fingers and PLA for the palm, arm, pulley, and other structural components.

### 2.2. Gesture Analysis of Human Hand Grasping

To provide clearer bionic goals for the design of dexterous hands, several studies have classified and analyzed different grasping operations according to their characteristics [24,25]. The GRASP taxonomy classifies and counts 33 gripping postures of the finger [26], as shown in Table 5. Its statistics can guide the structural design of dexterous hands. The utilization rate of thumb is as high as 97.7%, thus preserving the independent motion of the CMC joint. The movements of the four fingers other than the thumb are interrelated in half of the cases, while the fine grasping achieved by the third and second fingers, the frequency sum of both is as high as 45.3%. Therefore, two actuators are used to drive the four fingers to differentiate between precision-type grasping and power-type grasping. The final design of the present dexterous hand with the number of degrees of freedom and actuators is shown in Table 6.

### 2.3. Underactuation Structure Design

#### 2.3.1. Overall Drive Design

The tendon rope drive can greatly simplify the design complexity of the dexterous hand and can provide enough space for the pressure sensor to be mounted and alignment space. The overall transmission scheme of the dexterous hand is shown in Figure 2. The colored lines indicate the arrangement of the tendon ropes in the dexterous hand. The bending motion of the thumb is controlled by Servo 1, and a tendon rope (blue line) passes through the DIP and MCP joints of the thumb and connects directly to the output of Servo 1. The side-swinging motion of the thumb is controlled separately by servo 2, by a tendon rope (purple line) passing through the CMC joint of the thumb and connecting directly to the output of servo 2. The bending motion of the index and middle fingers is controlled by the servo 3 linkage, and the index and middle fingers are connected in series by a tendon rope (red line), and the roots of the two fingers pass through the differential mechanism in the palm, which balances the tension of the tendon ropes on the two fingers to realize the synergistic adaptive motion of the two fingers. The input of the differential mechanism is connected to the output of the servo 3 by another tendon rope (orange line). The bending motion of the ring finger and the little finger is linked and controlled by the servo 4. The ring finger and the little finger are connected in series by a tendon rope (green line), and the roots of the two fingers pass through another differential mechanism in the palm, which balances the tension of the tendon ropes on the two fingers to realize the synergistic adaptive motion of the two fingers. The input of the differential mechanism is connected by another tendon rope (cyan line) to the output of the servo 4.

#### 2.3.2. Finger Structure Design

In addition to the thumb, the four fingers have different external dimensions but share the same mechanical structure. This paper focuses on designing the finger to efficiently distribute sensors and manage their wiring within a confined space. The design also aims to prevent the wires from being torn during finger movement and to avoid any obstructions caused by finger flexing. To achieve this, the paper departs from the traditional shaft structure and instead employs a symmetric hemispherical head snap mechanism. This innovative design utilizes the elasticity of the materials to connect the knuckles, enabling a hinge structure that mimics a virtual “shaft” when the components fit into their designated slots. Furthermore, patch pressure sensors are used to maximize space at the fingertip [27,28]. The pressure sensor adopted in this paper is SBT760F and its performance is shown in Table 7. As shown in Figure 3. The slots on both sides of the joint are used to install torsion springs to achieve finger reset, and the slots are equipped with a stop structure to prevent the torsion springs from coming out of the slots, making the assembly easy. In the design of the thumb, the joint type is the same as the remaining four fingers, and only the degrees of freedom are different.

The parameter selection of the torsion spring is critical. If the initial torque is less than the gravitational torque of the finger, the finger will fail to reset completely. When the finger is bent, the driver must overcome the rebound force of the torsion spring to perform work. If the torque is too large, it will result in unnecessary energy consumption. Since the middle finger has the largest mass and the longest length, the gravitational torque experienced by its metacarpophalangeal (MCP) joint is the greatest. Therefore, the middle finger is taken as an example, and its MCP joint is considered the critical position. First, the gravitational force of the middle finger is estimated in Creo3.0 (a CAD design software package launched in October 2010 by the US company PTC), and the gravitational torque at the MCP joint is calculated to be approximately 1 N·mm. Based on this, the parameters of the torsion spring are selected, as shown in Table 8. The stiffness coefficient k of the torsion spring is given by the formula(1)$$k=\frac{E\times d^{4}}{3660\times n\times D_{m}}$$

The E is the modulus of elasticity, d is the wire diameter, n is the number of active coils, and $D_{m}$ is the mean coil diameter. As shown in the table, the elasticity coefficients and initial torque increments of the torsion springs at the three linkage joints—MCP, proximal interphalangeal (PIP), and distal interphalangeal (DIP)—are provided.

#### 2.3.3. Adaptive Differential Structure Design

In addition to the movement of a single finger, underactuation is also extended to multiple fingers, i.e., a single actuator is used to drive the movement of multiple fingers, again requiring a drive with adaptive characteristics. The driving force needs to be evenly distributed to each linked finger so that when the motion of one or more fingers is blocked, the remaining fingers continue to be driven. Introducing underactuation between fingers can further reduce the complexity of the system. Differential mechanisms that have been applied in dexterous hand research are moving pulleys [29], differential linkages [30], and differential gear [31]. In this paper, the moving pulley is used as a differential structure between fingers. The moving pulley has no limitation on the differential stroke, but the key rope is easy to lose and fall off in practical applications, so in this paper, a specific structure is used in the design to fix the trajectory of the moving pulley and to avoid the problem of the tendon rope falling off from the moving pulley, as shown in Figure 4. The pulley shell parameters are a = 0.35 mm, b = 1.1 mm; the effective travel of pulley A is c = 44 mm; and the effective travel of pulley B is d = 50 mm.

Based on the designed movable pulley structure, the force transfer sketch of a single movable pulley differential mechanism is established as shown in Figure 5, so as to obtain the relationship between the driving force and the tendon rope tension. The circle in the figure represents the moving pulley, which has two degrees of freedom of motion in the designed differential structure, which are translational motion along the pulley guide and rotational motion around the pulley axis. One end of the moving pulley is connected to the actuator, which is recorded as the input end, ignoring the friction, the input of the differential mechanism is the output force $F_{a}$ of the actuator. The other end of the moving pulley is recorded as the two outputs, which are connected to the two fingers, and the two output forces are $F_{1}^{a}$ and $F_{2}^{a}$. The radius of the moving pulley is *r*. $\alpha_{1}$, $\alpha_{2}$ denote the angle between the input and the output, respectively. h denotes the distance between the two tangent points of the tendon rope and the pulley at the output end, and according to the geometric relationship, we can obtain(2)$$h=r\left(\sin\alpha_{1}+\sin\alpha_{2}\right)$$

Remembering that the output quantity is $F_{o}$, the input quantity is $F_{i}$, and the force transfer matrix is $T^{f}$, the force transfer relationship between the input and output of the differential mechanism can be obtained as(3)$$F_{o}=T^{f}F_{i}$$

According to the geometrical relationship, we can obtain(4)$$F_{o}=\left[\begin{array}{c} F_{1}^{a}\\ F_{2}^{a} \end{array}\right],F_{i}=\left[\begin{array}{c} F_{a}\\ T_{s} \end{array}\right],T^{f}=\frac{1}{h}\left[\begin{array}{cc} r & \sin\alpha_{2}\\ r & -\sin\alpha_{2} \end{array}\right]$$
where $T_{s}$ represents the elastic moment of the system; the larger $T_{s}$ is, the more flexible the system is. In the case of no elastic element, $T_{s}$ is taken as 0. Substituting Equation (4) into Equation (3) gives the relationship between output and input as(5)$$F_{1}^{a}=F_{2}^{a}=\frac{F_{a}}{\sin\alpha_{1}+\sin\alpha_{2}}$$

From the above analysis, it can be seen that in the case of a fixed moving pulley trajectory, ignoring the effect of friction, the two outputs of the moving pulley differential mechanism are always equal. The relationship between input and output is determined by the tendon rope angle $\alpha_{1}$, $\alpha_{2}$, which is non-linear. Figure 6 shows a surface plot of the dynamic pulley transmission ratio $F_{i}^{a}/\;F_{a}$ versus the tendon rope clamp $\alpha_{1}$, $\alpha_{2}$.

According to Figure 4c, it can be seen that the positions of the two fingers driven by the same servo are symmetrical with the center of the pulley guide as the axis of symmetry, so that the moving pulley moves along the guide to any position $\alpha_{1}$, $\alpha_{2}$ is always equal. The relationship between the driving force provided by the actuator and the tension of the finger tendon rope can be further simplified:(6)$$\frac{F_{i}^{a}}{F_{a}}=\frac{1}{2\sin\alpha_{i}}$$

According to Equation (5), The relationship betweenthe dynamic pulley transmission ratio $F_{i}^{a}/\;F_{a}$ and the tendon rope clamp angle $\alpha_{i}$ are shown in Figure 7.

#### 2.3.4. Wrist and Forearm Design

A wrist flexion joint is designed at the wrist as shown in Figure 8. The wrist joint is driven by a linear motor using a linkage drive, and the power is transmitted through a simple crank-rocker mechanism, which converts the reciprocating linear motion of the linear motor into the bending and straightening motions of the wrist, respectively. The forearm is designed to reduce the size as the main design principle, the distance from its bottom to the wrist rotation axis is 110 mm, the width is 61 mm, the height is 51 mm, and the installation of the servo and the linear motor is shown in Figure 9. The Servo model is the GDW CLS, RS0707-270°, and Table 9 shows the dimensions and performance of this servo. The servo weighs 20 g, provides 270° of turning angle and 0.75 N·m of torque, and is small in size and low in cost.

### 2.4. Control System Design

#### 2.4.1. Construction of the Control System

The dexterous hand circuit system is mainly composed of four systems: ADC processing system, power management system, motor drive system, and communication system. The control system shown in Figure 10, where the black realization indicates the power supply relationship and the blue dotted line indicates the communication relationship between the modules. The ADC processing system amplifies, operates, converts, and streams the differential signals from the sensors into the multiplexer in the form of IIC data streams. This method can improve the sensor’s anti-interference ability to a greater extent, and also makes the motor drive compatible with the sensor’s long-wire transmission. Power management system adopts power matching mode for distributed management. Intelligent voltage reduction chips are used to effectively distribute the specific energy consumption, voltage, and current required by each system to achieve differentiated management of power and energy consumption. The linear motor drive system used to control the wrist motion utilizes a full H-bridge drive circuit that provides precise closed-loop control of the drive’s direction, speed, position, and current through pulse width modulation.

#### 2.4.2. The PID Control Based on Pressure Sensor

Since the sensors are subjected to pressure generated by the structure itself after assembly, and the effects vary depending on the installation, the sensors mounted in the fingers need to be calibrated. Vertical pressure is applied to the tip of the index finger using weights of 1 kg, 1.2 kg, 1.4 kg, and 1.6 kg, respectively. The three-byte hexadecimal data read by the upper computer is then converted into decimal numbers. The calibration curve and its fitting curve are shown in Figure 11.

The grasping operation of the underactuated humanoid dexterous hand is controlled in two stages: inverse kinematics-based open-loop position control and pressure sensor-based PID force feedback control. First, the coordinates of the fingertip are obtained through the vision system, either by pre-grasping or based on the shape of the target object. These coordinates are then used in the position control stage, where the desired servo rotation angle is calculated according to the diameter of the servo disk, and the servo is controlled to rotate to the desired position. Next, the system determines whether to enter the force control stage by comparing the detected pressure sensor value with a threshold value τ. Since precise closed-loop position control cannot be achieved to ensure the fingers reach the predetermined coordinates exactly, an error compensation mechanism is incorporated into the control process. If the fingertip does not touch the target object in the previous stage (i.e., the detected pressure sensor value is less than the threshold τ), the error compensation mechanism is triggered to increase the servo output. This process continues until the fingertip touches the target object and the detected pressure sensor value exceeds the threshold τ, at which point the pressure sensor-based PID force feedback control stage is activated.

In the force feedback control stage, a desired pressure is manually set based on the quality and flexibility of the target object. The classical PID control outputs a PWM wave to adjust the servo angle until the fingertip’s output pressure matches the desired pressure. In the PID closed-loop control, the force feedback control trigger threshold τ for the thumb tip pressure sensor is set to 0.05 N, and the expectation pressure is set to 0.4 N. During the error compensation stage, the thumb increases its bending angle incrementally. When the sensor detects a value greater than 0.05 N upon contacting the object’s surface, the PID closed-loop control is triggered. In this experiment, the maximum response frequency of the pressure sensor is 90 Hz, and the maximum response frequency of the servo is 50 Hz. To ensure the servo operates properly, the experiment’s operating frequency is set to 40 Hz. The proportional, integral, and derivative terms of the PID force–feedback closed-loop control strategy are selected as kp  = 27, Ti = 1, and Td = 0.8, respectively.

## 3. Motion Analysis of Underactuation Dexterous Hand

### 3.1. Driver Key Rope Mapping Relationships

The bending angle of each finger depends on a servo, the output of which is the length of the tendon rope. In touching the outside money, its turning angle has a definite geometric relationship. Taking the index finger as an example, the structure of the index finger is shown in Figure 12, and the relationship expression of the tendon displacement Δl is given for the three joint turning angles $\theta_{0},{\;\theta }_{1},{\;\theta }_{2}$:(7)$$\begin{array}{c} \Delta l=\left(b_{0}+b_{1}+b_{2}\right)-2c_{0}\sin\frac{\theta_{0max}-\theta_{0}}{2}\\ -2c_{1}\sin\frac{\theta_{1max}-\theta_{1}}{2}-2c_{2}\sin\frac{\theta_{2max}-\theta_{2}}{2} \end{array}$$
where $b_{i}$ denotes the length of the exposed tendon rope between the two knuckles in the natural state, $c_{i}$ denotes the distance between the center of rotation of the joint and the point of contact between the tendon rope and the knuckle, and $\theta_{imax}(i=0,\;1,\;2)$ denotes the maximum bending angle of the joint.

### 3.2. Dexterous Fingertip Trajectory Analysis

Due to the larger number of drivers in the thumb and the simpler coupling motion between the joints, only the index finger was selected as the object of study, and the remaining three fingers had the same structure and similar trajectory morphology as the index finger.

Figure 13 shows the static force equilibrium diagram of the index finger in the bending state, where $H_{i}$ is the force arm of the tendon rope tension for each joint, which can be obtained according to the geometric relationship:(8)$$\left\{\begin{array}{l} H_{2}=a_{2}\\ H_{1}={\;a}_{2}+{\;a}_{1}\cos\theta_{2}\\ H_{0}={\;a}_{2}+a_{1}\cos\theta_{2}+a_{0}\cos\left(\theta_{1}+\theta_{2}\right) \end{array}\right.$$

In the moving pulley differential mechanism, the relationship between the driving force Fa and the tension force $F_{1}^{a}$ applied to the tendon rope, ignoring the friction force applied to the tendon rope in motion, the tendon rope tension can be regarded as equal everywhere, i.e., $F_{1}^{a}=F_{\;1}^{’a}$. With the integral method to analyze the force of each joint separately, according to the balance of moments at the joints, $k_{i}\left(20{}^{\circ}+\theta_{i}\right)=F_{1}^{a}H_{i}$, the critical conditions for the rotation of the DIP, PIP, and MCP joints can be obtained as follows, respectively.(9)$$\left\{\begin{array}{l} k_{0}\left(20{}^{\circ}+\theta_{0}\right)=F_{1}^{a}\left(a_{2}+a_{1}+a_{0}\right)\\ k_{1}\left(20{}^{\circ}+\theta_{1}\right)=F_{1}^{a}\left(a_{2}+a_{1}\right)\\ k_{2}\left(20{}^{\circ}+\theta_{2}\right)=F_{1}^{a}(a_{2}) \end{array}\right.$$

During the movement of the finger, the three joints do not move together, but there is a sequence according to the critical conditions. In the torsion spring stiffness selection, take $k_{0}<k_{1}<{\;k}_{2}$. The calculation and values of ki(i=0, 1, 2) is given in Table 8. It is known that the initial angle of the torsion spring is 20°; in the equilibrium state, according to the moment applied by the tendon rope to each joint, it is equal to the torsion produced by the torsion spring; it can be seen that in the process of driving the finger to bend, when the tendon rope tension $F_{1}^{a}>\frac{k_{0}(20{}^{\circ}+\theta_{0})}{a_{2}+a_{1}+a_{0}}$, the MCP joints start to rotate, and both the PIP and the DIP joints remain stationary; when the MCP joint rotates to $\theta_{0}=\frac{20{}^{\circ}k_{1}(a_{2}+a_{1}+a_{0})}{k_{0}(a_{2}+a_{1})}-20{}^{\circ}$, the PIP joint starts to rotate, and the DIP joint remains stationary; when the PIP joint rotates to $\theta_{1}=\frac{20{}^{\circ}k_{2}(a_{2}+{\;a}_{1})}{k_{1}a_{2}}-20{}^{\circ}$, the DIP joint rotates to $\theta_{0}=\frac{20{}^{\circ}k_{2}(a_{2}+a_{1}+a_{0}\cos\theta_{1})}{k_{0}a_{2}}-20{}^{\circ}$, and the DIP joint starts to rotate.

According to the movement law of the finger, we can find out the relationship between the movement of each joint angle of the finger $\theta_{0},\theta_{1},{\;\theta }_{2}$ at different stages.

When $0\leq {\;\theta }_{0}<\frac{20{}^{\circ}k_{1}\left(a_{2}+a_{1}+a_{0}\right)}{k_{0}\left(a_{2}+a_{1}\right)}-20{}^{\circ}$,(10)$$\left\{\begin{array}{l} \theta_{2}=0\\ \theta_{1}=0\\ \theta_{0}=\theta_{0max}-2\arcsin\frac{b_{0}-\Delta l}{2c_{0}} \end{array}\right.$$

When $\frac{20{}^{\circ}k_{1}(a_{2}+a_{1}+{\;a}_{0})}{k_{0}(a_{2}+{\;a}_{1})}-20{}^{\circ}\leq \theta_{0}<\frac{20{}^{\circ}k_{2}\left(a_{2}+a_{1}+a_{0}\cos\theta_{1}\right)}{k_{0}a_{2}}-20{}^{\circ}$,(11)$$\left\{\begin{array}{l} \theta_{2}=0\\ \theta_{0}=\frac{k_{1}\left(\theta_{1}+20{}^{\circ}\right)\left(a_{2}+a_{1}+a_{0}\cos\theta_{1}\right)}{k_{0}\left(a_{2}+a_{1}\right)}-20{}^{\circ}\\ c_{0}\sin\frac{\theta_{0max}-\theta_{0}}{2}-c_{1}\sin\frac{\theta_{1max}-\theta_{1}}{2}=\frac{\left(b_{0}+b_{1}\right)-\Delta l}{2} \end{array}\right.$$

When $\frac{20{}^{\circ}k_{2}(a_{2}+a_{1}+a_{0}\cos\theta_{1})}{k_{0}a_{2}}-20{}^{\circ}\leq {\;\theta }_{0}\leq 80{}^{\circ}$,(12)$$\left\{\begin{array}{l} \theta_{1}=\frac{k_{2}\left(\theta_{2}+20{}^{\circ}\right)\left(a_{2}+a_{1}\cos\theta_{2}\right)}{k_{1}a_{2}}-20{}^{\circ}\\ \theta_{0}=\frac{k_{2}\left(\theta_{2}+20{}^{\circ}\right)\left(a_{2}+a_{1}\cos\theta_{2}+a_{0}\cos\theta_{1}\right)}{k_{0}a_{2}}-20{}^{\circ}\\ c_{0}\sin\frac{\theta_{0max}-\theta_{0}}{2}-c_{1}\sin\frac{\theta_{1max}-\theta_{1}}{2}-\;c_{2}\sin\frac{\theta_{2max}-\theta_{2}}{2}\\ =\frac{\left(b_{0}+b_{1}+b_{3}\right)-\Delta l}{2} \end{array}\right.$$

The relationship between the obtained joint angles $\theta_{0},{\;\theta }_{1},{\;\theta }_{2}$ was used to plot the motion trajectory of the index finger tip in Matlab software as shown in the red curve in Figure 12. The accuracy of the theoretical path is verified by the path measurement experiment, where a single index finger is fixed vertically on the profile, and the tendon rope is pulled by the motor with a speed of 3 mm/s in the direction of gravity at a constant speed, and the motion process of the index finger is photographed with a camera and randomly taken as a point. The coordinates of each point are recorded, and the motion path of the index finger is plotted as shown by the blue curves in Figure 14. The deviation between the experimental trajectory and the theoretical trajectory in the figure is mainly caused by the frictional resistance of the tendon rope in motion, resulting in an unstable motion relationship between the three joints, in which the largest error point is about 3 mm, which is 12.5% of the width of the finger, and can be ignored.

## 4. Grasping Ability Test

### 4.1. Grasping Posture Test

The grasping posture test method is as follows: the dexterous hand is fixed vertically, and the target object is manually moved to a suitable position. The dexterous hand is then driven to grasp the target object using the corresponding grasping posture. If the grasp is maintained stably for 10 s, it is considered a successful grasp.

The grasping postures of the dexterous hand prototype for objects of varying shapes and sizes are illustrated in Figure 15. The tested objects include an orange, an egg, a pen, a crisp cone, a plastic bottle, an eyeglasses case, a medicine bottle, double-sided adhesive tape, a ruler, and a student card. The dexterous hand successfully achieved 20 grasping postures categorized by the GRASP taxonomy, with a total usage frequency of 66%. These include power grasps for larger objects utilizing all five fingers, as shown in Figure 15a,d–f, where the sequence of finger movements for power grasping is detailed in Figure 15a. Precision pinching of smaller objects was performed using the tips of the thumb, index finger, and middle finger, as depicted in Figure 15b,g,i,j, with Figure 15b showcasing the sequence for precision pinching. Additionally, slender objects were pinched between the index and middle fingers, as shown in Figure 15c, while sheet-type objects were stably grasped by the tip of the thumb and the side of the index finger, as illustrated in Figure 15k. Out of the 13 postures where grasping failed, 10 instances were caused by the mechanical linkage between the index and middle fingers, 2 were due to the linkage between the MCP and PIP joints of the four fingers, and 1 failure resulted from insufficient surface friction, causing the object to slip. Despite these limitations, the majority of daily-life objects could still be successfully grasped using alternative postures. Overall, the underactuated dexterous hand designed in this study exhibits sufficient flexibility to adapt to most grasping tasks encountered in everyday applications.

In order to further demonstrate the working ability of the underactuation dexterous hand, objects in different environments are grasped in the stationary state of the target object. The characteristics and advantages of the underactuation dexterous hand are analyzed based on the dynamic working process. Since the size of the dexterous hand designed in this paper is similar to that of a human hand, the dexterous hand is fitted with dispensing labor gloves to increase the friction coefficient of the finger surfaces to improve the grasping performance, as shown in Figure 16. Figure 16a records the process of grasping and releasing a tennis ball by the dexterous hand, and the adaptive grasping process of the fingers according to the contour of the target object is visible in the figure. Figure 16b records the grasping process of the dexterous hand on a peach, in which the presence of an obstacle near the target object is visible, but due to the flexible joints and adaptive motion of the dexterous hand, the obstacle did not have an effect on the grasping of the target object. Figure 16c records the grasping process of the dexterous hand on the lid, due to the specific shape of the lid cannot accommodate more than one finger, five-finger or four-finger linkage of the hand cannot be grasped, while the dexterous hand designed in this paper except the thumb of the four fingers can be driven two by two individually, so it can be grasped on this type of object. Figure 16d records the grasping process of the dexterous hand on the student card, for the thin sheet-type object, the contact area of the finger side pinch is larger compared to that of the fingertip pinch, so the finger side pinch can achieve a more stable grasping effect, while the thumb of the dexterous hand designed in this paper has two driving degrees of freedom, which can achieve the independent bending of the thumb. In the figure, it can be seen that the bending force of the thumb can fall on the side of the index finger, which can stabilize the pinching of the thin sheet-like objects. Figure 16e records the process of the dexterous hand’s use of the spray bottle. In the figure, it can be seen that the dexterous hand firstly grasps the spray bottle stably through the thumb, ring finger, and little finger, and then drives the index finger and middle finger to bend to pinch the handle of the spray bottle, which is a skillful way to complete the use of the spray bottle. It can be seen that, because the dexterous hand designed in this paper has four degrees of freedom of drive, it can change the sequence of the four degrees of freedom of drive to achieve more complex operations, and has a greater potential for application.

### 4.2. Grip Strength Test

#### 4.2.1. Passive Grip Strength Test

The four fingers of the dexterous hand were driven to bend to appropriate positions. While the servos were enabled, bags containing weights of 200 g, 300 g, 400 g, and 500 g were hung on the fingers to simulate the posture of a human hand lifting a heavy object. Each test was maintained for 30 s to evaluate the dexterous hand’s ability to grip a 500 g weight under practical conditions. As shown in Figure 17, the dexterous hand demonstrated stable performance, successfully lifting the 500 g weight without any functional issues.

#### 4.2.2. Active Grip Test

Active grasping platform is shown in Figure 18, simulating the principle of grip strength meter. The dexterous hand end is fixed at one end of the two aluminum profiles. The other end of the profile is placed in the digital tensile tester. The slider on the profile blocks the tensile tester’s shoulders. The head of the tensile tester is tied to the end of the fishing line, and the other end of the line is tied to the pull rod. The pull rod falls naturally on the four fingers under the effect of gravity. To control the PWM, the four fingers gradually bend and adaptively hook the pull rod under the action of the differential mechanism to apply force to the palm direction. The position of the slider is adjusted so that the force gauge is moved back and the experiment is repeated until the slider is moved back to the point where the fingers cannot hold the lever. For the maximum PWM, the maximum active grip force of 4.8 N. Due to the limited length of the fingers, it can be verified that the dexterous hand can provide an active grip force with a maximum of 4.8 N under this experimental platform. Since testing the grip force limit would destroy the motor, no additional load experiment was conducted to test its maximum grip force while meeting the design requirements.

#### 4.2.3. Grip Ratio Test

The grip-to-weight ratio represents the ratio of the weight of the object grasped vertically by the dexterous hand to the weight of the dexterous hand itself and is an important measure of the dexterous hand’s grasping ability and structural robustness. As shown in Figure 19, repeated weight grasping tests with different positions of the bottle body as the grasping points have successfully grasped 500 g weight without relative displacement, which meets the design goal. The underactuation dexterous hand designed in this paper has a total weight of 0.35 kg plus the self-weight of the forearm, and it can be seen that its gripping weight ratio reaches 1.43 kg.

## 5. Conclusions

This study focuses on the underactuated dexterous hand, proposing a novel four-driven underactuated design that incorporates a moving pulley as the adaptive differential mechanism between fingers. The structural design effectively addresses the issue of tendon rope detachment from the moving pulley. Experiments demonstrate that the proposed mechanism achieves excellent performance in grasping postures, accommodating both power and precision grips. Moreover, it satisfies the target requirements in grip strength and grip weight ratio tests, thereby validating the feasibility of the design.

Despite meeting the fundamental requirements, several areas for improvement were identified during the design and experimental process. For instance, optimizing the stiffness of the torsion springs on each finger could enhance the finger’s movement trajectory under no-load conditions, thereby improving the dexterous hand’s grasping performance. Furthermore, a multi-dimensional, high-speed, and modular multi-layer controller could be developed to enable hierarchical task scheduling for the dexterous hand. Such a controller would facilitate the creation of a simulation system capable of accurately replicating the behavior of a fully actuated dexterous hand. Additionally, a safe and user-friendly human–computer interaction framework could be implemented, complete with an intuitive interface for achieving seamless online control of the underactuated dexterous hand.

## Figures and Tables

**Figure 1 biomimetics-10-00155-f001:**
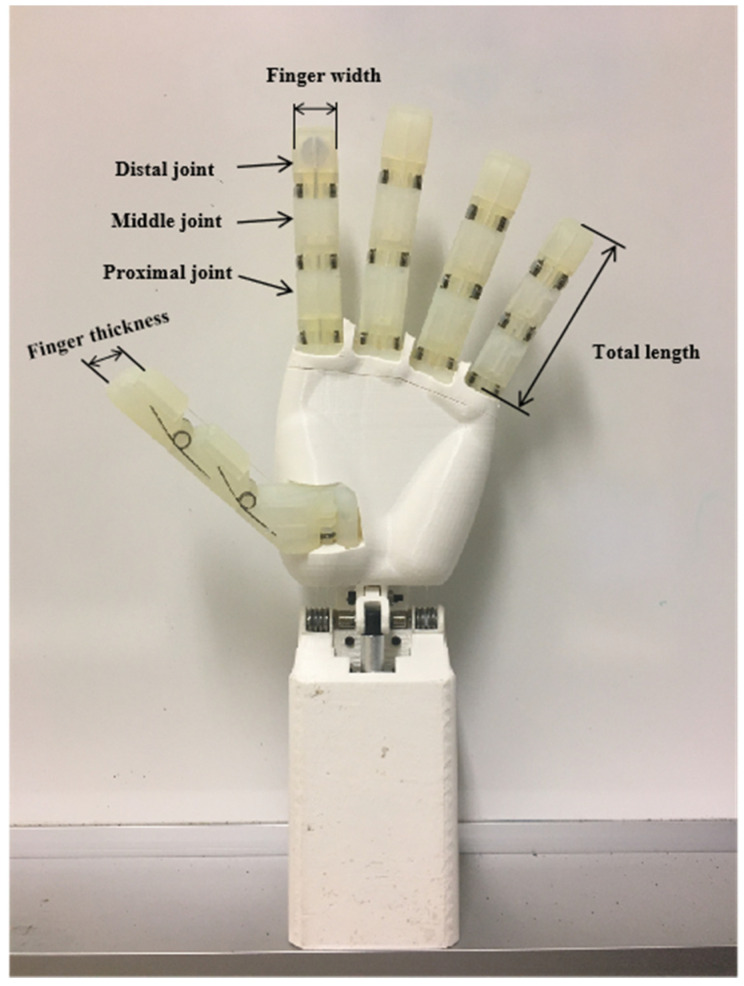
Scheme of the underactuated humanoid hand.

**Figure 2 biomimetics-10-00155-f002:**
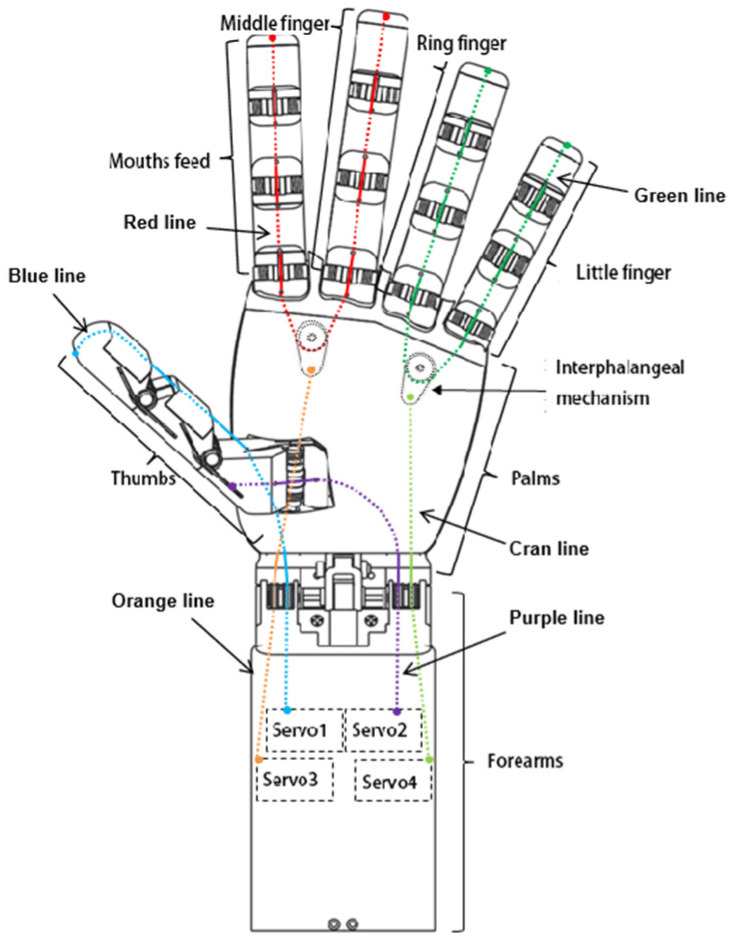
Modules and tendons distribution of the underactuated humanoid hand.

**Figure 3 biomimetics-10-00155-f003:**
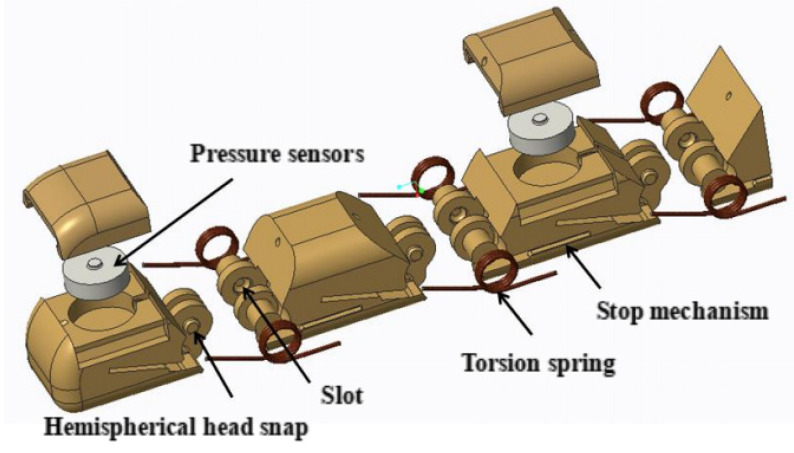
Exploded view of the finger model.

**Figure 4 biomimetics-10-00155-f004:**
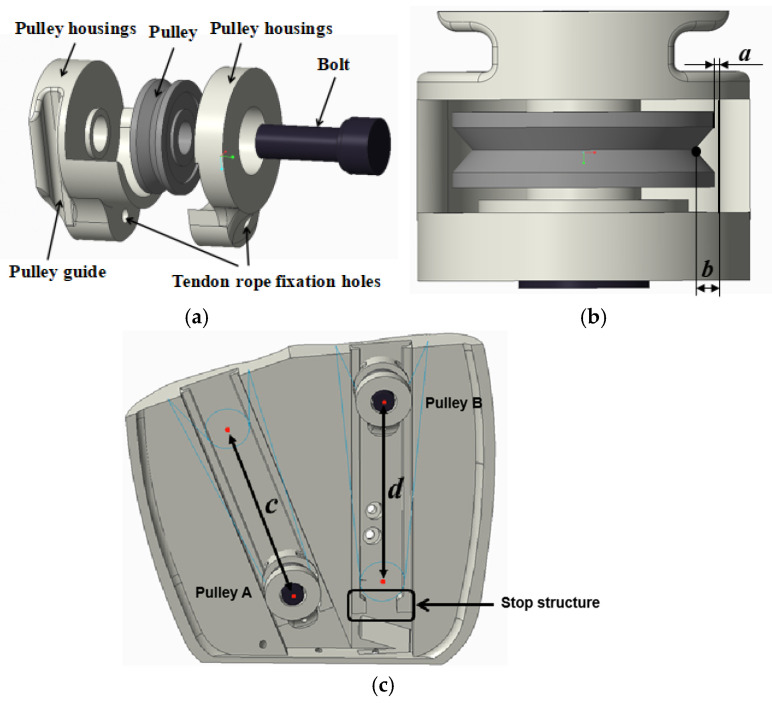
Design of moving pulley. (**a**) Exploded view of the moving pulley model. (**b**) Top view of assembled moving pulley. (**c**) Motion trajectory of the pulley in the palm.

**Figure 5 biomimetics-10-00155-f005:**
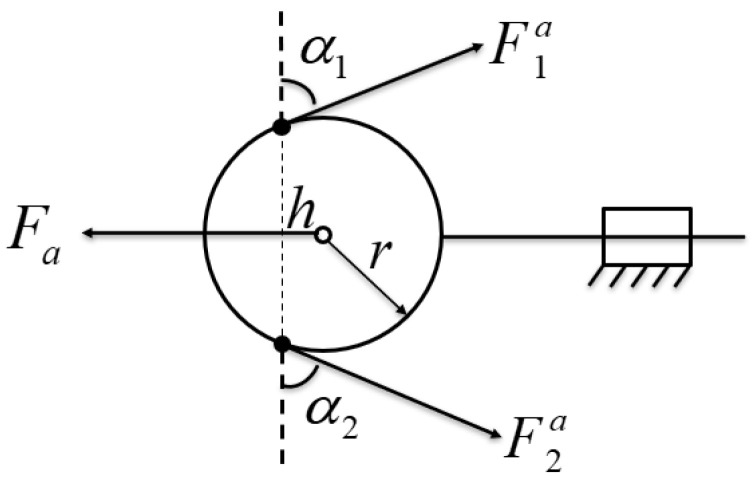
Modeling of the movable pulley.

**Figure 6 biomimetics-10-00155-f006:**
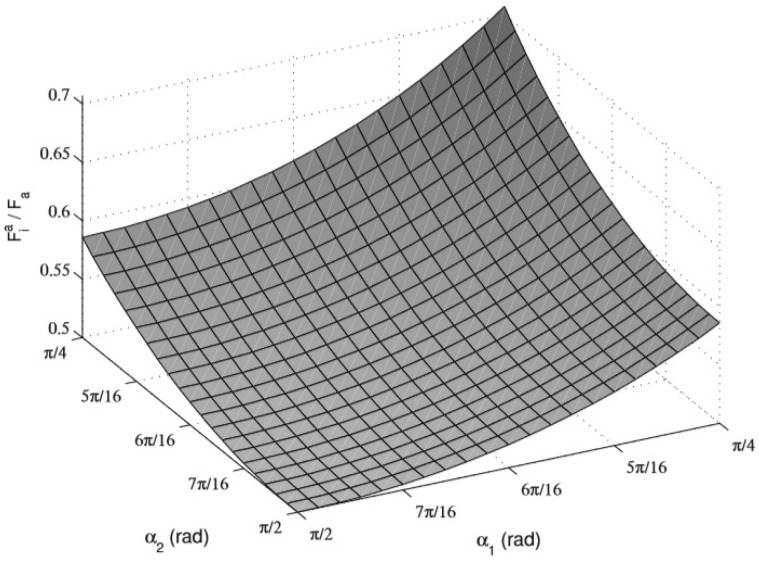
The universal relationship between force transmission ratio and tendon angles for movable pulley.

**Figure 7 biomimetics-10-00155-f007:**
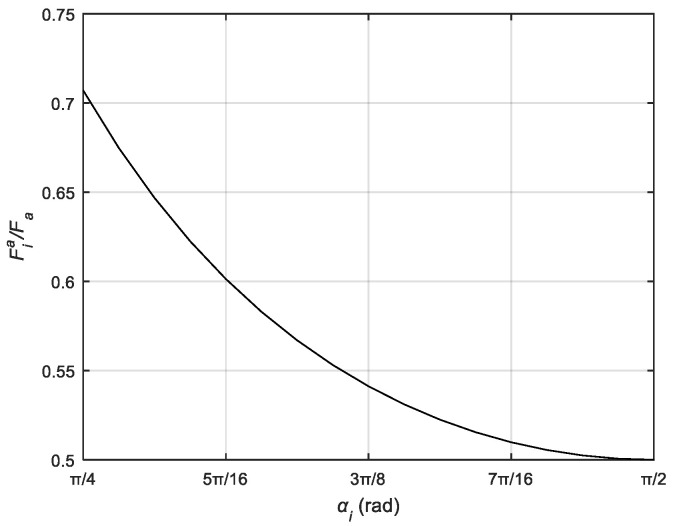
The relationship between force transmission ratio and symmetrical tendon angles for movable pulley.

**Figure 8 biomimetics-10-00155-f008:**
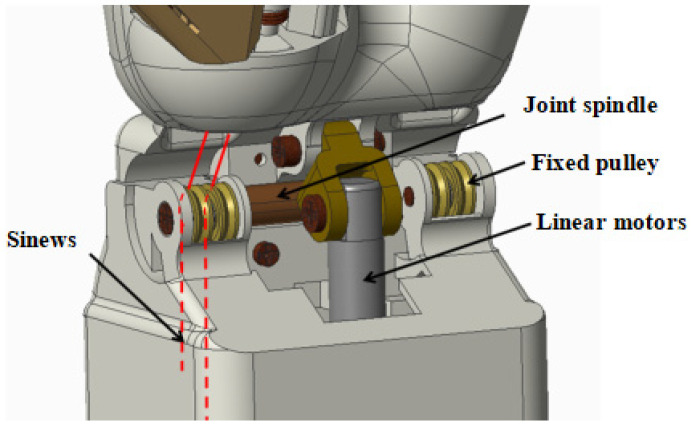
Structure design of the wrist.

**Figure 9 biomimetics-10-00155-f009:**
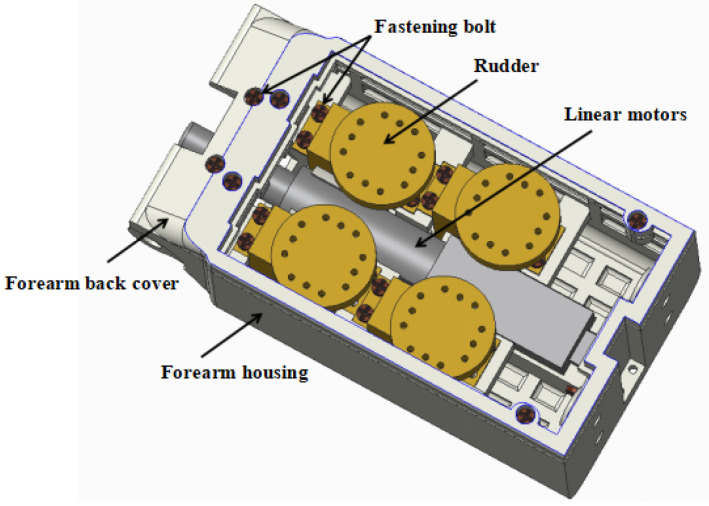
Structure design of the forearm.

**Figure 10 biomimetics-10-00155-f010:**
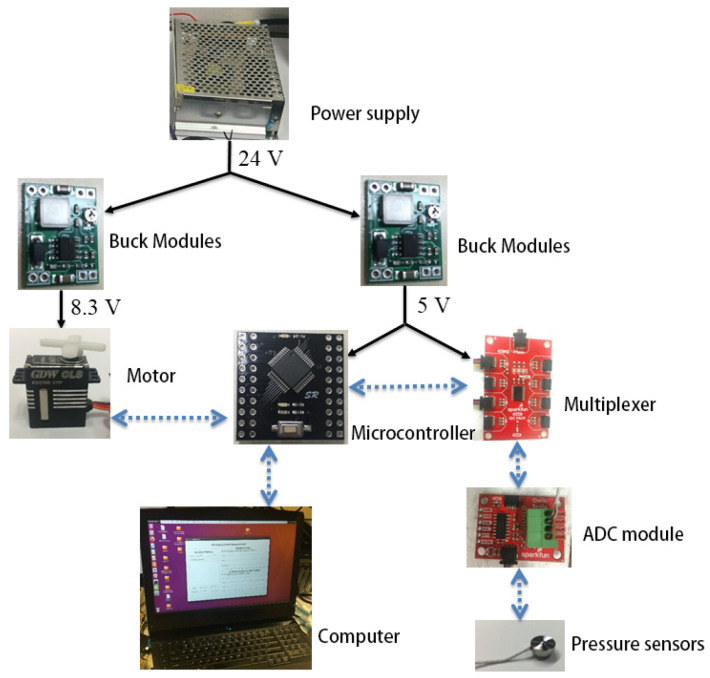
The outline of the circuit.

**Figure 11 biomimetics-10-00155-f011:**
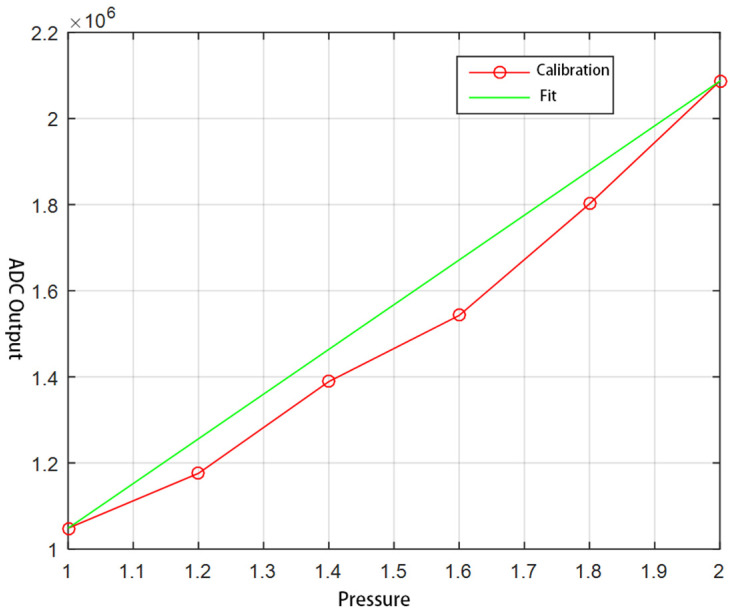
Calibration curve and fitting curve of force sensor.

**Figure 12 biomimetics-10-00155-f012:**
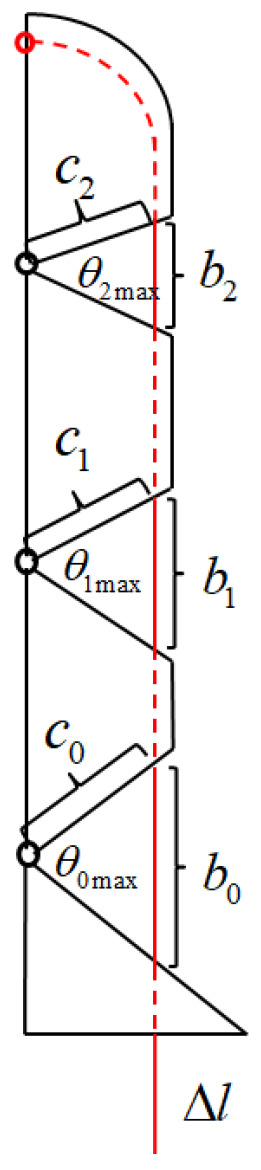
Geometric diagram of the index finger.

**Figure 13 biomimetics-10-00155-f013:**
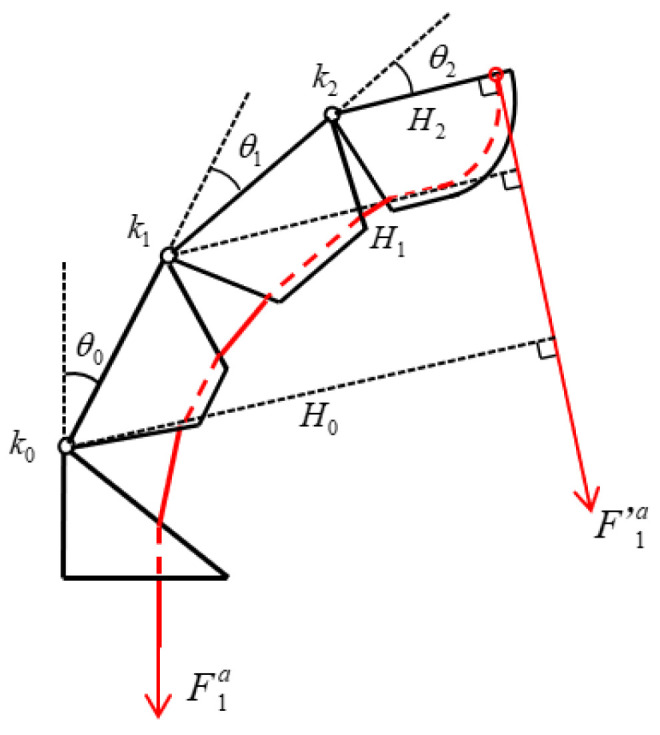
Static equilibrium diagram of the index finger in a bent state.

**Figure 14 biomimetics-10-00155-f014:**
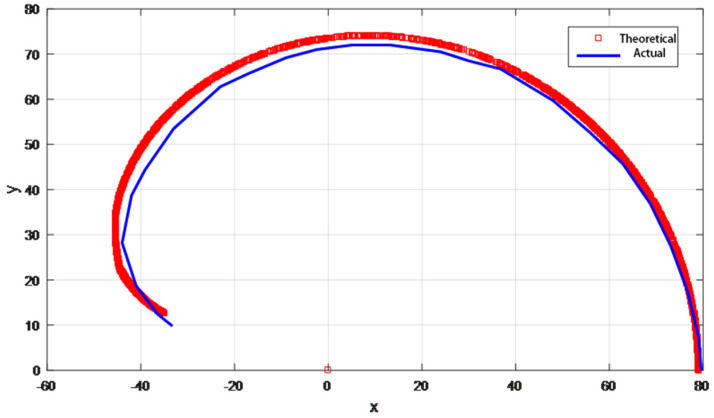
Motion trajectory of index fingertip.

**Figure 15 biomimetics-10-00155-f015:**
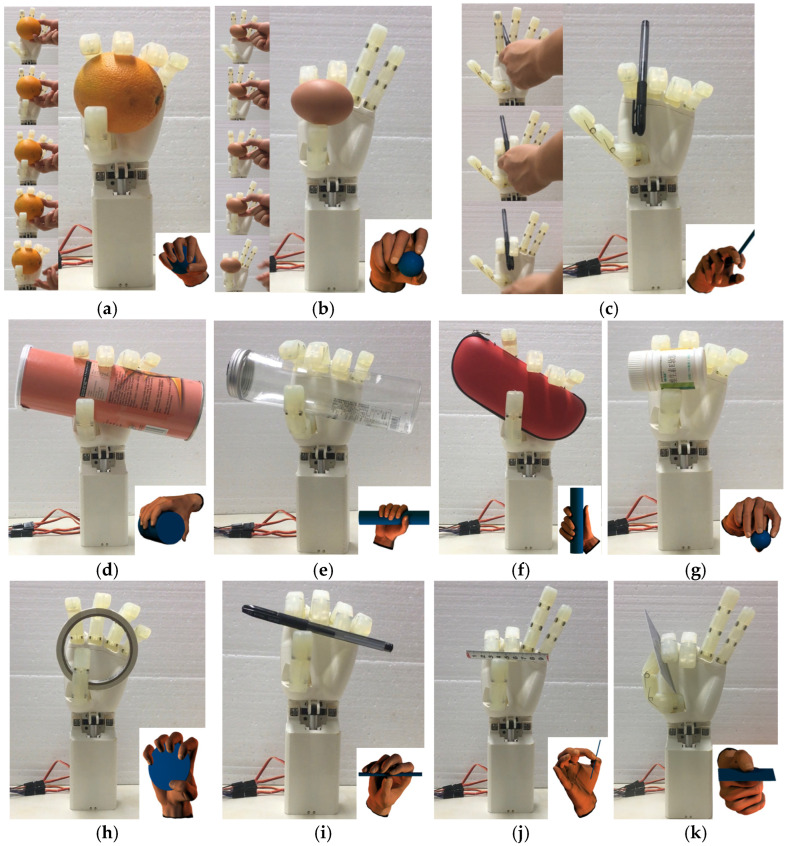
Test of grasping posture. (**a**) Orange. (**b**) Egg. (**c**) Pen. (**d**) Tube. (**e**) Plastic bottle. (**f**) Spectacle case. (**g**) Medicine bottle. (**h**) Adhesive tape. (**i**) Pen. (**j**) Ruler. (**k**) Student card.

**Figure 16 biomimetics-10-00155-f016:**
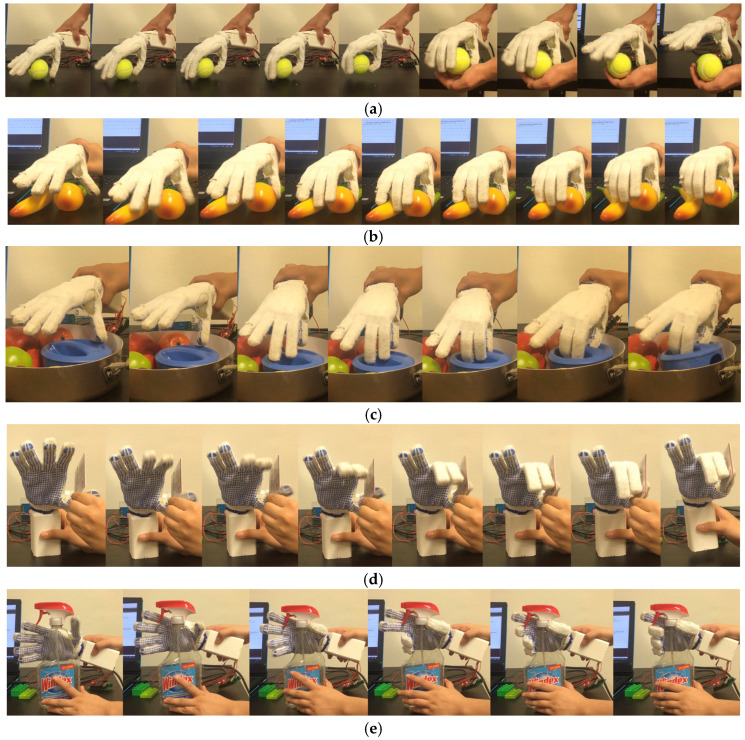
Grasping process. (**a**) Lifting tennis. (**b**) Lifting peach. (**c**) Lifting shell. (**d**) Lifting card. (**e**) Lifting sprinkling bottle.

**Figure 17 biomimetics-10-00155-f017:**
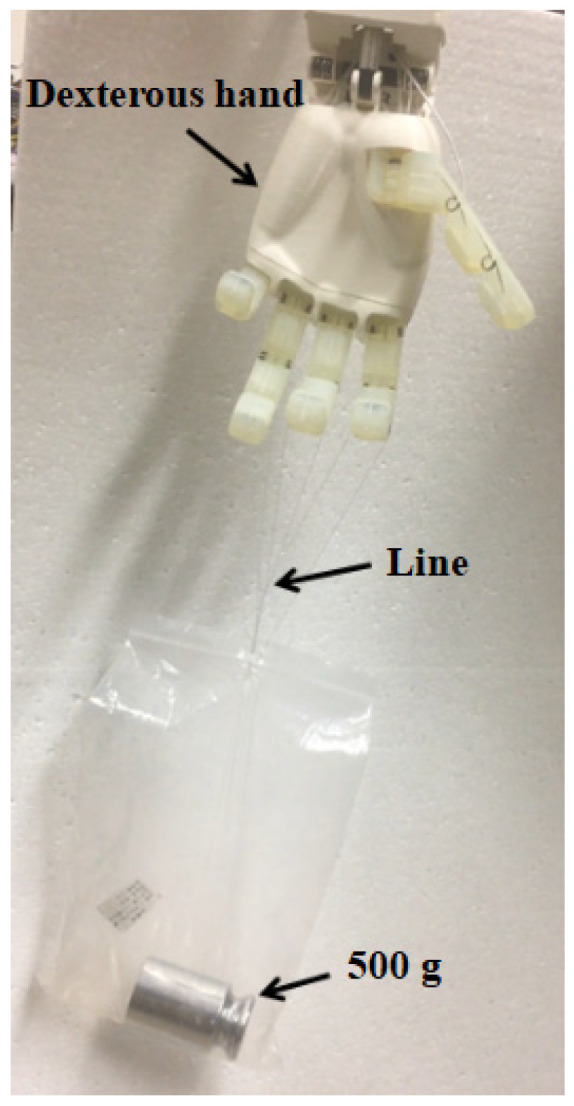
Test of passive grasping force.

**Figure 18 biomimetics-10-00155-f018:**
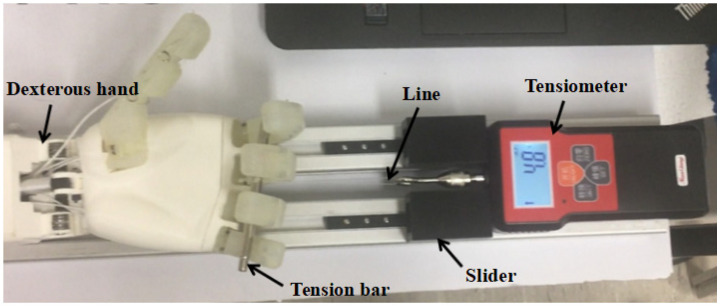
Test of active grasping force.

**Figure 19 biomimetics-10-00155-f019:**
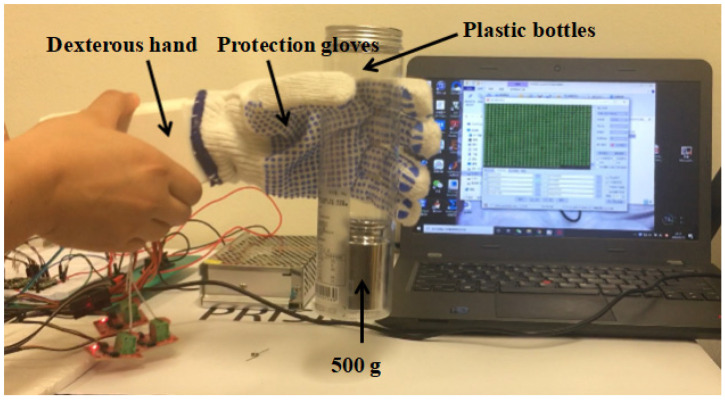
Test of grab–weight Ratio.

**Table 1 biomimetics-10-00155-t001:** Comparison of human hand sizes and prototype.

Dimensions (mm)	Average Length	Average Width	Average Hand Circumference
Adult male hand	193	89	218
Adult female hand	173	79	178
Dexterous hand	180	82	202

**Table 2 biomimetics-10-00155-t002:** Length of each knuckle and width and thickness of fingers.

Dimensions (mm)	Thumb	Mouths Feed	Middle Finger	Ring Finger	Little Finger
Distal joint	28	24	27	24	20
Middle joint	-	26	29	26	22
Proximal joint	31	29	32	29	25
Total length	59	79	88	79	67
Finger width	19	17	17	17	15
Finger thickness	16	16	16	16	14

**Table 3 biomimetics-10-00155-t003:** Active motion range of each joint.

	Thumb	Mouths Feed	Middle Finger	Ring Finger	Little Finger
DIP (°)	0–70				
PIP (°)	-	0–85			
MCP (°)	0–90				
CMC (°)	0–100	-	-	-	-

**Table 4 biomimetics-10-00155-t004:** Comparison of material prototype.

Material	Molding Method	Tensile Modulus (GPa)	Yield Elongation (%)	Heat Deflection Temperature at 0.45 MPa (°C)
PLA	FDM	2.34	3.3	56.0
General Resin	SLA	2.80	6.2	58.4
Wear-resistant resin	SLA	1.26	49.0	43.3

**Table 5 biomimetics-10-00155-t005:** Frequency statistics of gripping types based on GRASP taxonomy.

Contact Area	Thumbs Up for Participation	Adductor Hallucis Longus	Thumbs Participation	Five-Finger Participation	Four-Finger Participation	Three-Finger Participation	Two-Finger Participation	Power-Based	Precision	Intermediate
intermediate	31.4%	10.2%	21.6%	31.4%	0.4%	0%	0%	31.8%	0%	0%
facet	44.7%	2.0%	42.7%	17.1%	4.5%	16.3%	6.8%	7.3%	37.4%	0%
digit side	21.6%	20.9%	2.9%	1.6%	0%	0.7%	21.5%	0%	0.7%	23.1%
total frequency	97.7%	33.1%	67.2%	50.1%	4.9%	17%	28.3%	39.1%	38.1%	23.1%

**Table 6 biomimetics-10-00155-t006:** Freedom setting and drive assignment of dexterous hand.

	Tom Thumb	Mouths Feed	Middle Finger	Ring Finger	Little Finger	Finesse	Total
Number of bending degrees of freedom	2	3	3	3	3	1	15
Number of degrees of freedom for side pendulum	1	0					1
Number of drives	2	1		1		1	5

**Table 7 biomimetics-10-00155-t007:** Performance of SBT760F force sensor.

Range	Output Sensitivity	Zero Output	Nolinear	Repeatability	Hysteresis	Creep
10 kg	1.0~2.0 ± 10 mv/v	±0.2% F.S	0.5–1% F.S	0.3% F.S	0.3% F.S	0.25% F.S

**Table 8 biomimetics-10-00155-t008:** Parameters of torsion spring.

	DIP	PIP	MCP	CMC
Material	SUS304WPB			
E ($N/{mm}^{2}$)	186,000	186,000	186,000	186,000
d (mm)	0.7	0.65	0.6	0.6
Number of laps n	3	3	3	3
$D_{m}$ (mm)	5.3	5.4/5.3	5.4	5.4
Preload angle (°)	20	20	20	20
Number of torsion spring s	2	1/1	2	2
Coefficient of elasticity k	0.76	0.56	0.41	0.41
Initial torque (N·mm)	0.531	0.391	0.286	0.286

**Table 9 biomimetics-10-00155-t009:** Performance of RS0707-270° servo motor.

Input Voltage (V)	6.0	7.4	8.4
Speed (s/60°)	0.12	0.1	0.08
Torsion (kg·cm)	6.0	6.8	7.5
Operating current (mA)	400	500	600
Plugging current (mA)	1200	1350	1500
Weights (g)	20		

## Data Availability

The original contributions presented in this study are included in the article. Further inquiries can be directed to the corresponding author.

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
