# Peer review of "Structural and Experimental Study of a Multi-Finger Synergistic Adaptive Humanoid Dexterous Hand"

_biomimetics, 2025, doi:10.3390/biomimetics10030155_

Round 1
Reviewer 1 Report
Comments and Suggestions for Authors
Dear authors, thank you for this very interesting paper. Structural and experimental study of a multi-finger synergistic adaptive humanoid dexterous hand is very interesting the work.
This work presents the dexterous hand where finds to replicate human hand functionality where the article introduces the design and development of a multi-finger synergistic adaptive humanoid dexterous hand with underactuation flexible articulated fingers.
There are some recommendations to improve the quality of this contribution. Thank you for sharing this work and knowledge.
I recommend that the article be accepted but first it is necessary to correct it. Here are my comments:
1. It is necessary that all the manuscript must be revised and corrected because there are some grammatical errors.
2. Include a diagram of the position representation according to table 2 is shown.
3. Give the characteristics of the patch-type sensor that is used. Include the information in the body of the article.
4. Figure 5c does not exist. Correct, it is figure 4c
5. A list of materials with which each part of the prototype is built must be included and an explanation given.
6. In the paragraph “In the moving pulley differential mechanism, the relationship between the driving force Fa and the tension force 𝐹 1𝑎 applied to the tendon rope, ignoring the friction force applied to the tendon rope in motion. ¿What is the reason to ignore the friction?
7. From line 291, Figure 13 is not correct. Please check this part.
8. What would be the surface conditions for it to have a friction surface and not fail when picking up objects? Give an explanation and include in the body article.
9. An extensive section of the article must be included where the electrical control and detection of manipulation of the dexterous hand are explained in a general way. Explain it and include all the control parameters.
Comments on the Quality of English LanguageThe English needs to be reviewed, there are some spelling mistakes.
Reviewer 2 Report
Comments and Suggestions for Authors
Authors in this paper proposed the design and development of a multi-finger synergistic adaptive humanoid dexterous hand with underactuation flexible articulated fingers and integrated pressure sensors. Paper is well writed and the topic is very interesting, however to improve it:
1.- In page 2 line 62 Change "Finally, The.. " by "Finally, the.."
2.- In page 2, line 72, in the literature the human finger has 4 DOF, for the index and the middle, for the ring and small fingers there are more than 4 DOF to arch the palm, and the thumb has almost 5 DOF or more depend of the author.
3.- In page 3, Table 2, the sume of total length for the middle finger not is correct 86, the sume is 88.
4.- In page 4, Table 4, the first column looks like a two rows and the rest of table are four rows, can you arrange?
5.- In page 5, Figure 4(b), what are the horitzontal distance a and b? Could the authors define it?
6.- In page 9, line 244, there are theta_2max, add theta_omax and theta_1 max or put theta_imax, where i=o,1,2
7.- In page 10, line 263, change "F_a" by Fa.
8.- In page 10, line 266, define ki.
9.- In page 10, line 270, "the three joints do not move together", in human hand the joint move coupling, the DIP is difficult to move along relative to the PIP.
Round 2
Reviewer 1 Report
Comments and Suggestions for Authors
Dear authors, thank you for this very interesting paper. Structural and experimental study of a multi-finger synergistic adaptive humanoid dexterous hand is very interesting the work.
The authors have provided the information that is necessary to improve the article, but there are some observations and recommendations.
Thank you for sharing this work and knowledge.
I recommend that the article be accepted but first it is necessary to correct it the minor observations.
Here are my comments:
- Regarding the new information that was included, the author used some grammatical contractions, that in general is not considered correct.
- There are double Table 8, please verify what is correct it. Also, the author must correct in the manuscript the sequence.
- Regarding the new information that was included in the manuscript, please verify the grammatical.
Comments on the Quality of English Language
The authors need to check the grammar again.
Author Response
Manuscript ID biomimetics-3465047
Structural and experimental study of a multi-finger synergistic adaptive humanoid dexterous hand
Responses to Reviewer 1
The authors would like to express their sincere appreciation to the reviewer for his/her constructive comments and suggestions, and his/her time and efforts spent in helping us to improve the quality and presentation of the paper.
- Comment: “Regarding the new information that was included, the author used some grammatical contractions, that in general is not considered correct.”
Response: Thank you for your comment and suggestion. We've given an explanation of the abbreviation.
Creo is a CAD design software package launched in October 2010 by the US company PTC.
- Comment: “There are double Table 8, please verify what is correct it. Also, the author must correct in the manuscript the sequence.”
Response: Thank you for your comment and suggestion. This error has been corrected. In the meantime, we've cut out two unnecessary figures and have given the correct numbering.
- Comment: “Regarding the new information that was included in the manuscript, please verify the grammatical.”
Response: Thank you for your valuable suggestion. We've changed the grammar on a full-scope basis.